# The Dynamic Changes in Biosynthesis and Spatiotemporal Distribution of Phytohormones Under Jasmonic Acid Treatment Provide Insights into Hormonal Regulation in *Sinopodophyllum hexandrum*

**DOI:** 10.3390/plants14071001

**Published:** 2025-03-22

**Authors:** Siyu Shen, Yuqing Wu, Yunfeng Luo, Yang Li, Wei Gao, Luqi Huang, Yating Hu, Kang Chen, Yuru Tong

**Affiliations:** 1School of Traditional Chinese Medicine, Capital Medical University, Beijing 100069, China; dooriyashen97@ccmu.edu.cn (S.S.); luoyunfeng@westlake.edu.cn (Y.L.); weigao@ccmu.edu.cn (W.G.); 2State Key Laboratory for Quality Ensurance and Sustainable Use of Dao-di Herbs, National Resource Center for Chinese Materia Medica, China Academy of Chinese Medical Sciences, Beijing 100700, China; huangluqi01@126.com; 3School of Pharmaceutical Sciences, Capital Medical University, Beijing 100069, China; wuyq992021@163.com (Y.W.); liyang001020@163.com (Y.L.); 4State Key Laboratory for Quality Ensurance and Sustainable Use of Dao-di Herbs, Beijing 100700, China

**Keywords:** *Sinopodophyllum hexandrum*, jasmonic acid, transcription factors, hormonal regulation

## Abstract

*Sinopodophyllum hexandrum* (Royle) Ying, the only species of Sinopodophyllum in *Berberidaceae*, is an endangered traditional Tibetan medicine. The harsh plateau growth environment makes *S. hexandrum* tough to breed and meet the global demand for clinical medications such as podophyllotoxin (PTOX) and etoposide. Jasmonic acid (JA) is acknowledged as a key phytohormone that modulates stress responses by activating defense mechanisms and promoting the production of specialized metabolites, which offers valuable insights for developing varieties that are more resilient to stress or yield higher amounts of secondary metabolites. In this study, JA treatment was used as a simulated source of stress to investigate the spatiotemporal changes in phytohormones, such as JA, *cis*-(+)-12-oxo-10, 15(*Z*)-phytodienoic acid (*cis*-(+)-OPDA), and abscisic acid (ABA), and transcriptional regulation following hormonal regulation in intact plants. Some correlations through changes in phytohormone levels and the expression level of related signaling pathway genes were observed to confirm the overall regulatory effect after the JA treatment. Furthermore, the JA treatment caused the differential expression of various genes including transcription factors (TFs), of which the most typical one is myelocytomatosis oncogene like protein 2 (MYC2), *ShMYC2_3*. Therefore, we proposed that a plant hormone-mediated regulatory network exists endogenously in *S. hexandrum*, enabling it to respond to JA treatment. This study provides a new direction for the germplasm improvement and the sustainable utilization of *S. hexandrum* when facing exogenous stimulation.

## 1. Introduction

*Sinopodophyllum hexandrum* (Royle) Ying, a perennial herb of podophyllaceae (Berberidaceae), has been regarded as a precious wild medicinal resource due to its harsh growth environment [1,2] and long harvest period (at least 4 years for the plant to mature) [3]. Its dried roots and rhizomes are a famous traditional Chinese medicine named *taoerqi* (Sinopodophylli Radix et Rhizome), which was first recorded in the Tibetan Medical Classic *Yuewangyaozhen* and has effects of treating rheumatism, promoting blood circulation, relieving pain, and reducing expectoration. The main bioactive ingredients of *S. hexandrum* are flavonoids and aryltetralin-type lignans such as PTOX, which is the chemical semi-synthetic precursor of etoposide, a commonly used drug for conditions such as germ cell tumors, small-cell lung cancer, and non-Hodgkin’s lymphoma [4]. According to the data released by the China Pharmaceutical Industry Research Institute, the global etoposide market has exceeded USD 1 billion, and is still growing. However, at present, *S. hexandrum* has become a national grade II rare and endangered plant in China due to the slow renewal of its wild population. Meanwhile, owing to the serious destruction of wild germplasm resources, low natural germination rate of seeds, limitation of high-altitude planting land, and long growth cycle (4–6 years) [3,5,6], the industrialization and in-depth development and utilization of *S. hexandrum* are obstructed. Therefore, it is necessary to study the endogenous regulation process under adverse stress conditions of *S. hexandrum* to provide more information on the growth and development of plants and the optimization of stress-resistant germplasm.

Phytohormones are natural organic compounds that play crucial roles in physiological processes, including plant growth, development, stress responses, and metabolite regulation [7,8,9]. As one of the major phytohormones, JA and its derivatives such as methyl jasmonate (MeJA) and jasmonoyl-isoleucine (JA-Ile), collectively referred to as jasmonates (JAs), mediate plant responses to environmental stress and are involved in various developmental processes, including root growth, seed germination, pollen development, and fruit development and maturation, serving as signaling molecules [10,11,12,13]. JA treatment can optimize plant physiological performance through complex signaling networks and interactions with other hormones. For example, in Arabidopsis, the application of JA can induce the expression of resistance-related genes (such as *PDF1.2*), thereby enhancing the plant’s defense against insects and pathogens [14]. In maize, JA treatment can promote the release of volatile organic compounds (such as terpenoids and green leaf volatiles), attracting natural enemies or parasitoid wasps, and enhancing the plant’s indirect defense capabilities [15]. Zhu et al. found that exogenous JA treatment enhances the salt tolerance of wheat by regulating the expression of biosynthetic genes for phytohormones such as ABA and salicylic acid (SA) [16]. Meanwhile, JAs exert regulatory effects on nearly all secondary metabolites, including terpenoids [17,18], alkaloids, and lignans, which serve as the pharmacological basis for many clinical drugs. For instance, MeJA upregulated the biosynthesis of immunomodulating triterpenoid saponins, such as ginsenosides, through MeJA-responsive TFs like MYB2, WRKY8, and WRKY9 in *Panax ginseng* [19,20,21]. JA signaling-mediated TFs, including WRKY, bHLH, ERF, TCP, and YABBY, have been identified as promoters of artemisinin biosynthesis in *Artemisia annua* [22]. Furthermore, the spatiotemporal expression analysis suggests that JA may be an important signaling molecule regulating saponin biosynthesis in *Panax notoginseng* [23].

However, most studies on the JA-mediated regulation of *S. hexandrum* have mainly focused on suspension cells [24,25]. In contrast, few have investigated the regulatory mechanisms of phytohormones under JA treatment in intact plants, which can more accurately reflect the plant’s intrinsic regulatory state. Additionally, previous research has largely overlooked the internal spatiotemporal dynamics of phytohormones and their associated regulatory pathways in *S. hexandrum*, leading to a significant gap in the understanding of the influence and crosstalk of phytohormones in these plants. Therefore, analyzing the dynamic changes in the phytohormone content and transcriptional regulatory trends in *S. hexandrum* under JA treatment, which mimics stress signals, will provide valuable insights into the defense responses and regulation of secondary metabolites in this medicinal plant. In this study, we sprayed 100 μM JA aqueous solution on the aerial parts of 3-year-old *S. hexandrum* plants and determined the spatiotemporal content changes and gene expression level changes in three endogenous phytohormones (OPDA, JA, and ABA) in the stems and roots of the samples, with different application times. Transcriptome sequencing was also conducted on each sample to uncover the transcriptional regulation and crosstalk between phytohormones, revealing a precise and rapid regulatory network mediated by phytohormones in response to environmental stress. These findings offer valuable insights into the regulation of phytohormones, which will contribute to germplasm improvement and the development of new stress-resistant *S. hexandrum* varieties.

## 2. Results

### 2.1. Dynamic Response Characteristics of JA Pathway Under JA Treatment

Since JA plays an important role in plants’ response to mechanical damage or external stress, we have firstly paid attention to the changes in the JA content in different tissues after JA application. As shown in Figure 1b, we observed that the JA content in stems rapidly increased within the first 0–0.5 days after JA treatment, followed by a gradual decline. By analyzing the expression trends of genes involved in the JA biosynthetic pathway, we found that in stems, almost all the pathway genes showed an upward trend in expression levels shortly after JA application (within 0–3 days). This was especially notable for the genes related to the OPDA biosynthesis pathway, including *13-LOX_1/3/4/5*, *AOS*, and *AOC*, and those involved in the conversion of OPDA to JA, such as *OPR3*, *OPCL1,* and *ACX_2*. At the same time, however, the content of JA in the roots was too low to be detected, which corresponded to the fact that JA was mainly located in the aboveground part and the most in the flowers, as expected. The expression levels of most genes were low in the roots before and after JA application (Figure 1b,d, Appendix A).

OPDA is a key signaling molecule in plants’ response to complex environmental changes, playing a significant biological role in immunity, defense, development, and stress responses. However, when detecting the endogenous OPDA levels, despite significant variation within the groups, the overall trend showed a rapid decline in the content of endogenous OPDA (Figure 1a), opposite to the increasing expression levels of the OPDA biosynthetic pathway genes. We found that, in addition to the CORONATINE INSENSITIVE 1 (COI1)-dependent pathway, OPDA regulates various physiological and developmental processes in plants through multiple mechanisms. For example, OPDA activates signaling pathways by interacting with receptors or channels on the cell membrane, interacts with other phytohormones such as auxin and gibberellin, regulates transcription factors, and suppresses or activates specific genes [26]. Given the multiple COI1-independent functions of OPDA in plants, we hypothesized that the JA treatment not only influences the JA-COI1-related pathways but also activates OPDA-COI1-dependent bioactivity. Once OPDA was synthesized, it was likely rapidly metabolized or converted into other bioactive molecules to respond to the plant growth and defense mechanisms triggered by JA, thereby contributing to disease resistance, stress tolerance, and developmental processes, which could lead to a lower accumulation of OPDA in *S. hexandrum*.

### 2.2. The High Expression Level of JAR1 Mainly Contributes to the Rapid Decline in JA Levels, Which Transfers to JA-Ile and Rapidly Oxidizes to Become Inactivated

As it is transferred into the cytosol, JA is catalyzed by jasmonoyl-isoleucine synthetase (or jasmonate resistant 1, JAR1) to form JA-Ile, the most biologically active form among jasmonates, which mediates COI1-dependent core JA signaling [27]. The COI1 receptor detects JA-Ile and initiates its degradation by recruiting Jasmonate ZIM-domain (JAZ) inhibitors. This process releases downstream TFs, activating signaling pathways that regulate plant resistance, growth, and development [28]. These JA-responsive signal transductions will affect the endogenous secondary metabolism of plants and promote the expression of secondary metabolite biosynthetic pathway genes. However, in this study, the content of JA-Ile in both the stems and roots was too low to be detected. Therefore, we were unable to directly observe the impact of the JA treatment on the JA signaling pathway based solely on the content levels. Based on transcriptome data, we found that the initial expression level of *JAR1* in the roots was relatively high, indicating that JA in the roots could rapidly be converted to JA-Ile in the early stages. Additionally, COI1, which binds to JA-Ile, also showed high expression levels. These phenomena were consistent with the observation that endogenous JA in roots was barely detectable at the initial stage. Additionally, we observed that the expression level of *JAR1* in the stems increased on the third day after JA induction. This increase may explain the reduction in endogenous JA observed between 0.5 d and 3 d (Figure 1b–d, Appendix A).

Regarding the fate of JA-Ile, we speculated that after exerting its effects, JA-Ile is rapidly degraded and converted into inactive substances through oxidative metabolism. Oxidation at the C-12 position is a key pathway for the inactivation of JA-Ile, as the resulting oxidation products cannot facilitate the assembly of the COI1-JAZ co-receptor complex [29]. These oxidation processes are mainly catalyzed by multiple members of the cytochrome P450 94 family including CYP94B1, CYP94B3, and CYP94C1 [30,31,32]. By analyzing the expression levels of CYP94 family genes in the transcriptome, we found that the expression of *CYP94B1*, *CYP94B3*, and *CYP94C1_1* in the stems increased to varying degrees at 0.5 d and 3 d, showing a consistent pattern with the changes in the *JAR1* expression (Figure 1c,d, Appendix A). From these, we concluded that JA and its signaling pathways are rapid and transient.

### 2.3. The Synergistic Effect of JA and ABA in Response to JA Treatment

JAs rarely operate in isolation. Instead, they integrate into complex networks that interact with other phytohormonal signaling pathways. Previous studies have emphasized the intricate transcriptional regulatory network involved in artemisinin biosynthesis in A. annua, revealing significant interactions between JAs and ABA [33]. Upon measuring the change in the ABA content, we observed a subtle upward trend in the stem ABA levels on the third day after JA application. The transcriptome data indicated that several genes in the ABA biosynthesis pathway, including *ZEP_1/2/4* and *NCED* (highlighted in orange), were upregulated to varying degrees, which correspond to the observed increase in the ABA content (Figure 2, Appendix A). Interestingly, the JA administration had some impact on ABA production even in the roots, which means that the signaling is, to some extent, working in these organs, even if no increase in the JA content has been observed. Based on the changes in the JA content mentioned above, the endogenous ABA and JA responses to the JA treatment exhibited synergy. This may be related to the shared downstream effector molecules or signaling regulatory mechanisms, such as those involving TFs like MYC2. These common molecules enable the flexible modulation of both signaling pathways as plants respond to complex environmental conditions. Therefore, in the following section, we conducted a detailed analysis of TFs and metabolic networks.

### 2.4. ShMYC2_3 May Serve as a Key Regulator Within a Complex Network That Orchestrates the Plant’s Response to JA Signals While Mediating the Production of Secondary Metabolites in S. hexandrum

Given that phytohormone effects are typically short and rapid, we analyzed the differentially expressed genes (DEGs) in the stems after JA application at day 0 and day 3. The genes with an adjusted *p*-value < 0.05 and |log2FoldChange| > 1 were selected for further analysis. A total of 1978 DEGs were identified, with 1205 upregulated and 773 downregulated genes (Figure 3a, Appendix A). Based on the Kyoto Encyclopedia of Genes and Genomes (KEGG) analysis, a total of 771 DEGs were mapped to 231 metabolic pathways. Among these, significant gene differences were observed in pathways related to phytohormone signal transduction, flavonoid biosynthesis, plant–pathogen interactions, and plant stress defense. These include JA signaling pathway genes such as *JAR1* and *JAZ*, the key chalcone biosynthesis gene chalcone synthase (CHS), and TFs like *MYC2* and *WRKY33*. Genes involved in pathways such as isoflavonoid biosynthesis, α-linolenic acid metabolism, and phenylpropanoid biosynthesis and metabolism also showed differential expression shortly after JA application. Additionally, cutin, suberine, and wax biosynthesis refer to the processes through which plants synthesize and deposit protective layers on their surfaces. These compounds are crucial for maintaining plant integrity and responding to environmental stresses (Figure 3b, Appendix A).

We then investigated the most common TFs involved in regulation. Previous studies have identified four TFs—bZIP, MYB, WRKY, and bHLH—regulating secondary metabolite biosynthesis in two PTOX-producing plants, *S. hexandrum* and *Podophyllum peltatum*. Two unique transcripts encoding bHLH and MYB/SANT TFs were identified in the shoots of *P. peltatum*, while bZIP and MYB TFs were found in the rhizomes of *S. hexandrum*, based on differential expression levels [34]. To provide insights for transcriptional regulation in the JA treatment, we conducted an analysis on the JA induction and differential expression of TFs in the transcriptomes. The expression levels of various *MYB*, *bZIP*, *WRKY*, and *bHLH* in the roots and stems over the 0–3-day period exhibited significant differences. As shown in Figure 4 and Appendix A, the TFs that were highly expressed in the stems were generally low or not expressed in the roots (*MYB_24-42, bZIP_1-10, WRKY_5-17,* and *bHLH_1-30*), whereas the TFs with higher expression levels in the roots were found to be lower in the stems (*MYB_1-21, bZIP_13-21, WRKY_1-4,* and *bHLH_31-42*). Notably, we observed that the expression levels of some *MYB*, *bHLH, bZIP,* and *WRKY* TFs increased rapidly (indicated by the orange frame), suggesting that these TFs are highly sensitive to JA induction and may play a role in the biosynthesis of secondary metabolites.

MYC2 is a well-studied bHLH family TF in plants that acts as a transcriptional activator of JA signaling. It also plays a role in secondary metabolism and various growth and development processes. A previous study found that the biosynthesis of salvianolic acid B, along with the phenylalanine and tyrosine biosynthesis pathways, was significantly induced in 2-month-old transgenic *Salvia miltiorrhiza* plants that overexpressed *SmMYC2* [35]. The amino acid sequence of SmMYC2 (GenBank ID: AHN63211.1) was used as a reference to screen for *ShMYC* genes with a transcript length of more than 300 bp and a similarity of over 20% to *SmMYC2*. Five candidate genes that met these criteria were identified, and their expression levels in the stems and roots following JA application were analyzed. The results indicated that the expression levels of all five *ShMYC2* genes in the stems increased, with *ShMYC2_3* showing the most significant rise, reaching its highest level on the third day (Figure 5a, Appendix A). The phylogenetic analysis of these five candidate sequences in relation to *SmMYC2* revealed that *ShMYC2_3* clustered closely with *SmMYC2* (Figure 5b, Appendix A). Therefore, we speculate that *ShMYC2_3* is likely involved in the regulation of endogenous secondary metabolism in *S. hexandrum*. Further detailed experiments will be necessary to investigate this hypothesis.

## 3. Discussion

The high altitude, low temperature, and strong presence of ultraviolet light [1,2] of its growing environment and its long growth cycle (4–6 years) [3] make *S. hexandrum* greatly restricted by the plant growth cycle, cultivation conditions, and ecological environment, leading to issues with the available resources of this medical plant. JA is a phytohormone widely found in plants, which plays an important role in regulating the growth and development of plants. At present, most of the related studies on JA-induced *S. hexandrum* are mainly based on suspension cells [24,25], while less attention has been paid to the study of hormonal regulation and transcriptional spatiotemporal regulation of intact plants. Therefore, the elucidation of the regulatory mechanism of JA treatment on 3-year-old *S. hexandrum* will be conducive to our understanding of the metabolic regulation of this rare and endangered species, and enable us to provide services for the establishment of high-quality *S. hexandrum.*

In this study, we first focused on the JA levels after JA treatment. We found that the content of JA in the stems rapidly increased. Although the increase in the JA content in the stems may be due to exogenous uptake or neosynthesis, we did find that most genes involved in JA biosynthesis were significantly upregulated. The subsequent rapid decline in the JA levels further supported the notion that JA exerted its role in plant hormonal regulation through dynamic response characterization, which is very fast and efficient.

The changes in OPDA, the key intermediate of JA biosynthesis, however, showed a complementary pattern to those of JA. Previous studies have shown that OPDA, the precursor of JA, could also play the role of a signal molecule in salt response and resistance to insect invasion [36,37]. Based on the experimental results and theoretical analysis, we speculated that the JA treatment not only activates the JA-COI1-related signaling pathway but also potentially triggers COI1-independent signaling pathways through OPDA. This, in turn, initiates a series of physiological responses, involving plant defense, stress adaptation, and immune responses, as well as growth and development. For example, a signal transduction route mediated by OPDA-Ile, which is independent of the SCFCOI1-JAZ co-receptor [38]. Therefore, OPDA plays a crucial role in JA-induced plant responses, and its mechanism is likely to be multilayered and involve multiple pathways, making it an important signaling molecule for plants to adapt to complex environmental changes and the stress response.

The high expression level of JAR1 may mainly contribute to the rapid decline of endogenous JA within 0.5–3 days. Unfortunately, we were unable to detect JA-Ile in both the roots and stems. Here, we speculated that the reaction involving JA-Ile is instantaneous, and it would degrade rapidly after exerting the activity, making the corresponding COI1 quickly return to its original low expression level. Since the first detection time is at 0.5 d, JA-Ile may have already lost its activity due to the oxidation activity of the CYP94 family genes. Supporting this, the expression level of *CYP94B1*, *CYP94B3,* and *CYP94C1_1* did increase in the stems. Previous studies have shown that CYP94B3 exerts negative feedback control on the JA-Ile levels and performs a key role in the attenuation of jasmonate responses [31]. Therefore, we believed that the increased expression levels of the oxidative metabolism genes were also part of feedback to maintain the homeostasis of endogenous hormones in *S. hexandrum.*

Also, JA transporters are key components of the JA signaling pathway, responsible for the transport of JA across cellular membranes, ensuring its effective distribution throughout the plant and activation of related signaling pathways. These transporters include ATP-binding cassette G-type transporters (ABCGs, such as AtABCG16 in *Arabidopsis*) [39], lipid transporters (LPTs, e.g., DIR1 [40]), and Multidrug and Toxic compound Extrusion (MATE) transporters [41]. These proteins transfer JA and its derivatives from stimulated cells to others, regulating intracellular JA levels and initiating both local and systemic defense responses. In this study, we also analyzed the expression levels of ABCGs, LPTs, and MATE in the transcriptome (Appendix A) and found that the expression levels of LPTs were specifically increased in response to JA induction, particularly in the roots. This suggested that although JA and its derivatives in the roots were not detected, there still existed a notable hormonal regulation process and crosstalk occurred in the plants.

Among other endogenous phytohormones, we selected ABA as the detection object to prove that endogenous phytohormones do not act independently, but play a role in a complex network that intersects with other plant hormone signaling pathways. Previous studies have shown that the synergistic interaction between JA and ABA may be an important way for JA signaling to respond to abiotic stresses [42]. There is a link between the core components of the ABA signaling pathway and the JA response [43]. For instance, GhTLP1 from *Gossypium hirsutum* responded to JA and ABA, regulating the mitogen-activated protein kinase (MAPK) signaling pathway to enhance resistance in *Arabidopsis* plants against *Verticillium dahliae* [44]. In this study, we found that the ABA content in the stems increased in a short time, which is consistent with the results of previous studies that JA and ABA have a synergistic effect [45,46]. Further studies are needed to reveal the more detailed mechanism.

The results of the KEGG metabolic pathway enrichment analysis of the DEGs showed that the genes with significant differences after the JA treatment were mainly enriched in plant hormone signal transduction, flavonoid biosynthesis, and plant–pathogen interaction pathways, indicating that JA treatment can promote the metabolism and regulation of endogenous phytohormones and cause crosstalk interaction between phytohormones. The specific mechanism remains to be further studied. Flavonoids are another class of effective ingredients in *S. hexandrum.* The significant difference in the flavonoid biosynthesis pathway further indicates that JA has a vital effect on the endogenous secondary metabolites of *S. hexandrum.* Previous studies have shown that external stimuli such as light intensity and UV-B irradiation can increase the accumulation of flavonoids in *S. hexandrum*, allowing *S. hexandrum* to adapt to an elevated altitude coupled with high light intensity [2,47]. In these results, the relative expression of CHS, which is involved in flavonoid biosynthesis, was enhanced in response to external stimuli compared to CK. This is consistent with the results of this study. At the same time, among the DEGs, we also screened some TFs such as *MYC2* and *WRKY33* from the transcriptome, which have tissue expression differences and induced expression differences and are worthy of further study. Here, we focused on *SmMYC2* and found five related *ShMYC2s*. The regulatory effect of *ShMYC2_*3 on secondary metabolism in *S. hexandrum* will be further verified through yeast two-hybrid experiments. For example, by combining the genes involved in lignan biosynthesis with ShMYC2_3, we will explore how ShMYC2_3 activates gene expression and regulates metabolic pathways through interactions with target genes, thereby revealing the physical interaction between ShMYC2_3 and the pathway genes.

In this study, we described the mechanism of the JA-mediated spatiotemporal regulation of endogenous phytohormones and transcriptional regulation, and speculated the potential regulatory process of secondary metabolism (Figure 6). In the follow-up study, we will pay more attention to the important elements of TFs in transcriptional regulation, and combine the content of secondary metabolites such as lignans and flavonoids to carry out related studies, with the aim of providing new insights for the endogenous metabolic regulation and transcriptional regulation of *S. hexandrum*, providing a scientific basis for the sustainable utilization of its resources.

## 4. Materials and Methods

### 4.1. Plant Materials and Sample Collection

Three-year-old *S. hexandrum* plants grown from seeds from Zhuoni County, Gannan Tibetan Autonomous Prefecture, China (102°46′–104°02′ E, 34°10′–35°10′ N, 2000–3000 m altitude, temperate continental climate, with an average annual temperature of 8–10 °C, an average relative humidity of 50%, and more than 2000 h of sunshine per year; the ultraviolet radiation is relatively strong), were selected and divided into six groups with three replicates in each group. The species was identified by professor Mengfei Li of Gansu Agricultural University. After the acclimation of plants collected from semi-wild cultivation for three days (to minimize stress and allowed the plants, which were collected from semi-wild cultivation, to adapt to the new environment), 0-day samples were collected. Subsequently, all aerial parts of plants were treated with 100 μM JA solution, diluted from 2.38 M JA ethanol stock solution, until small droplets formed on the leaves; the droplet size was about 0.035 mm to 0.049 mm. The potted plants were then covered with plastic film to create a sealed environment for 2 h. The spraying process was repeated on the aerial parts, followed by resealing with the plastic film for three times. After removing the plastic film, samples were collected at various time points (Appendix A).

Samples from five other time points (day 0.5, 3, 7, 14, and 21) of three-year-old *S. hexandrum* plant induced by JA were selected and three biological replicates were used for subsequent experiments. The stem and root parts of three-year-old *S. hexandrum* plants were separated and cleaned. After completely removing the surface water, samples were frozen by liquid nitrogen and stored at −80 °C.

### 4.2. Extraction of Phytohormones

The sample treatment method is based on previous research [48]. Samples from six time points and two plant parts were finely ground in a mortar. An exact weight of 0.50000 g from each quick-frozen sample was placed into a 15 mL centrifuge tube, with all procedures conducted under liquid nitrogen. To each tube, 5 mL of extraction solvent (2-propanol/H_2_O/concentrated HCl at a 2:1:0.002 ratio) was added, ensuring a sample-to-solvent ratio of 1:10 (g·mL^−1^). The tubes were shaken at 100 rpm for 30 min at 4 °C. Subsequently, 2 mL of dichloromethane was added, and shaking continued for another 30 min at the same temperature and speed. The samples were centrifuged at 13,000× *g* for 5 min at 4 °C, resulting in two distinct phases with plant debris located in between. The lower phase solvent (7200 µL) was carefully transferred to a new 15 mL tube and concentrated, but not completely dried, using a nitrogen generator. Finally, the concentrated samples (20 µL) were dissolved in 800 µL of 50% methanol, filtered through a 0.22 μm membrane, and stored at -20 °C for later quantification.

### 4.3. LC-MS Conditions for Phytohormone Detection and Standard Curve Plotting

Ultra-high electrostatic field orbital trap Fourier transform mass spectrometry (Q Exactive HF, ThermoFisher scientific, San Jose, CA, USA) was used for phytohormone determinations. A Waters ACQUITY UPLC HSS T3 analytical column (2.1 mm × 100 mm, 1.8 μm) and Waters ACQUITY UPLC HSS T3 VanGuard Pre-column (2.1 mm × 5 mm, 1.8 μm) were used in combination in this analysis. The flow rate was kept at 0.4 mL/min and the temperature of the column oven was set at 30 °C. The mobile phase was composed of 0.05% formic acid water (A) and acetonitrile (B). To separate the phytohormones in the samples, gradient elution was set as follows: 0 min at 10% B, 4 min at 60% B. The injection volume was 5 μL. Mass spectra were acquired in negative-ion mode over a scan range of *m*/*z* 50–1500 with a scan time of 0.2 s. The ESI and MS settings were as follows: capillary voltage, 1.0 kV; cone voltage, 40 V; source temperature, 120 °C; desolvation temperature, 450 °C; cone gas flow, 50 L·h^−1^; desolvation gas flow, 800 L·h^−1^. Xcalibur 4.2 Qual Browser was used for data analysis.

### 4.4. mRNA Sequencing and Library Construction

Total RNA was extracted from each tissue using a modified cetyltrimethylammonium bromide (CTAB) method. RNA purity was assessed with a Kaiao K5500^®^ Spectrophotometer (Kaiao, Beijing, China). RNA integrity and concentration were evaluated using the RNA Nano 6000 Assay Kit on the Bioanalyzer 2100 system (Agilent Technologies, CA, USA). Each sample utilized 2 μg of RNA as input material for library preparation. Sequencing libraries were constructed using the NEBNext^®^ Ultra™ RNA Library Prep Kit for Illumina^®^ (#E7530L, NEB, Ipswich, MA, USA) according to the manufacturer’s instructions, with index codes added to distinguish sequences from each sample.

In brief, mRNA was purified from total RNA using poly-T oligo-attached magnetic beads. Fragmentation was performed using divalent cations under elevated temperature in NEBNext First Strand Synthesis Reaction Buffer (5×). The first strand of cDNA was synthesized with random hexamer primers and RNase H. Subsequently, the second strand of cDNA was synthesized using buffer, dNTPs, DNA polymerase I, and RNase H. The library fragments were purified with QiaQuick PCR kits and eluted with EB buffer, followed by terminal repair, A-tailing, and adapter ligation. The desired products were retrieved, and PCR amplification was performed to complete the library.

The RNA concentration of the library was measured using the Qubit^®^ RNA Assay Kit on a Qubit^®^ 3.0 to obtain a preliminary quantification, then diluted to 1 ng/μL. The insert size was assessed with the Agilent Bioanalyzer 2100 system, and precise quantification of the qualified insert size was conducted using the StepOnePlus™ Real-Time PCR System (library valid concentration > 10 nM). Clustering of the index-coded samples was performed on a cBot cluster generation system using the HiSeq PE Cluster Kit v4-cBot-HS (Illumina) following the manufacturer’s guidelines. After cluster generation, the libraries were sequenced on an Illumina platform, generating 150 bp paired-end reads.

### 4.5. Quantification of Gene Expression Using FPKM in Transcriptome Sequencing

In this study, transcriptome sequencing was performed on all plant samples. The gene expression levels were quantified using the Fragments Per Kilobase of transcript per Million mapped reads (FPKM) method. This method normalizes the gene expression data by considering both the length of the gene and the sequencing depth, providing a reliable measure for comparing gene expression levels between different genes or samples. The FPKM calculation is as follows:$$\mathrm{FPKM}=\frac{Number\;of\;fragments\;mapped\;to\;a\;gene}{Gene\;length\;(in\;kilobases)\;\times \;Total\;number\;of\;mapped\;fragments\;(in\;millions)}$$

This standardization method enables accurate comparisons of gene expression across different samples and conditions, accounting for variations in both gene length and sequencing depth. FPKM values were used to assess differential gene expression and provide insight into the gene activity within different experimental treatments.

### 4.6. RNA-Seq Data Analysis, De Novo Transcriptome Assembly, and Annotation

Sequence filtering was conducted prior to assembling the transcript sequences. Raw reads were cleaned by eliminating sequencing adapters and low-quality bases. In the absence of a reference genome for this species, the Trinity pipeline was employed for de novo transcriptome assembly. To explore the potential functions of the assembled transcripts, unigene sequences were compared with public databases such as Gene Ontology (GO) and the KEGG. R clusterProfiler was then utilized to perform GO functional classification and enrichment analysis, enabling an assessment of gene function distribution. The transcripts were classified into three primary functional categories: cellular component (CC), molecular function (MF), and biological process (BP) (Appendix A).

### 4.7. Differential Gene Expression Analysis

Significantly differentially expressed genes were identified by analyzing read count data from gene expression quantification with DESeq2 4.1.1 software. Screening thresholds were established at an adjusted *p*-value < 0.05 and |log2 (fold change)| > 1. The fold changes in expression levels between samples served as the main criterion for selection. After merging the DEGs from the two treatments, clustering analyses were performed using Cluster version 2.1.0 (Appendix A).

## Figures and Tables

**Figure 1 plants-14-01001-f001:**
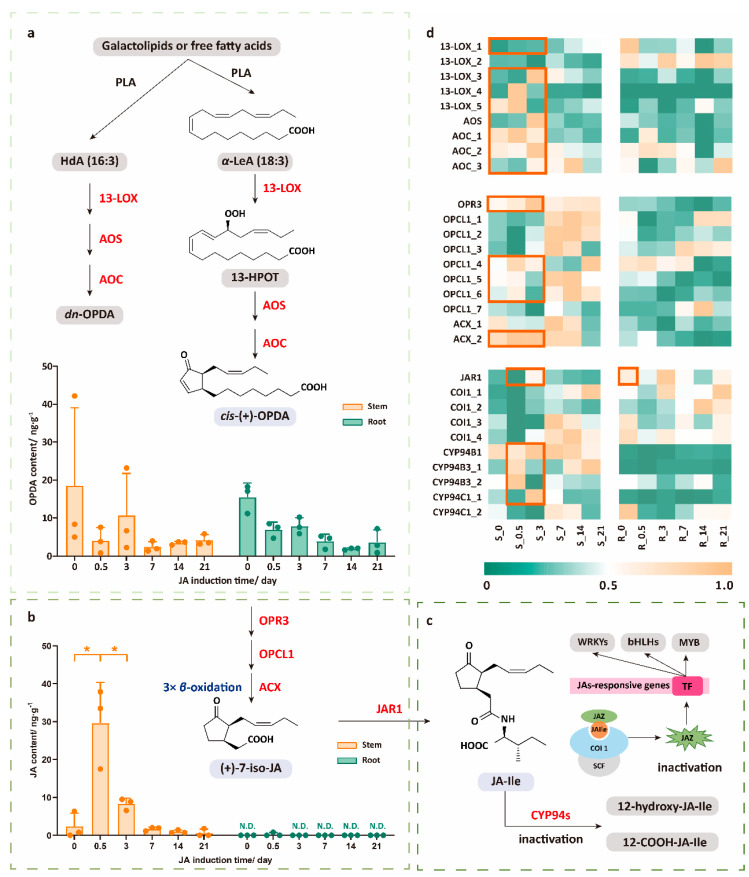
Biosynthetic and metabolic pathway of endogenous jasmonates. (**a**) The biosynthetic pathway and content change of OPDA; (**b**) the biosynthetic pathway and content change of JA; (**c**) the metabolic pathway of JA and the mechanism of JA-Ile; (**d**) expression pattern of JA biosynthetic and metabolic pathway genes. R stands for roots, while S stands for stems. α-LeA: α-linolenic acid, HdA: hexadecatrienoic acid, PLA: phospholipase, 13-LOX: 13-lipoxygenase, AOS: allene oxide synthase, AOC: allene oxide cyclase, OPR3: OPDA reductase 3, OPCL1: OPC-8:0 coenzyme A ligase1, ACX: acyl-CoA oxidase, JAR1: jasmonate resistant 1. The orange box represents genes with increased expression. * represents a significant difference between groups, *p* < 0.05.

**Figure 2 plants-14-01001-f002:**
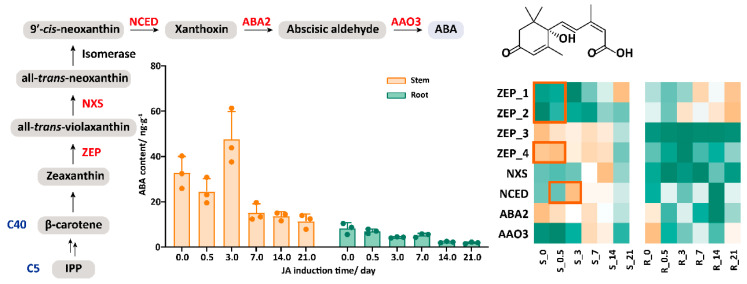
Content changes of endogenous ABA and expression pattern of ABA pathway genes. R stands for roots, while S stands for stems. ZEP: zeaxanthin epoxidase, NXS: neoxanthin synthase, NCED: 9-cis-epoxycarotenoid dioxygenase, ABA2: xanthoxin dehydrogenase, AAO3: abscisic aldehyde oxidase. The orange box represents genes with increased expression.

**Figure 3 plants-14-01001-f003:**
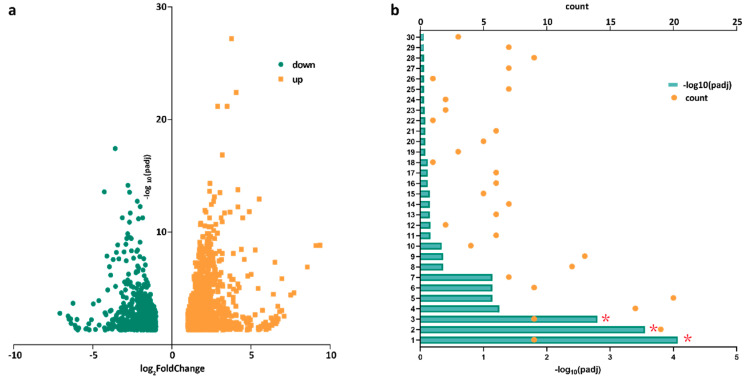
Analysis of differential genes in day 0 and day 3 stems. (**a**) All differential genes at 0 d vs. 3 d; (**b**) KEGG analysis. 1: Cutin, suberine, and wax biosynthesis. 2: Plant hormone signal transduction. 3: Flavonoid biosynthesis. 4: Plant–pathogen interaction. 5: Starch and sucrose metabolism. 6: alpha-Linolenic acid metabolism. 7: Circadian rhythm-plant. 8: Phenylpropanoid biosynthesis. 9: Amino sugar and nucleotide sugar metabolism. 10: Carotenoid biosynthesis. 11: Inositol phosphate metabolism. 12: Insulin secretion. 13: Photosynthesis. 14: Toll-like receptor signaling pathway. 15: Phenylalanine metabolism. 16: Leishmaniasis. 17: Cyanoamino acid metabolism. 18: Isoflavonoid biosynthesis. 19: Streptomycin biosynthesis. 20: Quorum sensing. 21: NF-kappa B signaling pathway. 22: Primary immunodeficiency. 23: Linoleic acid metabolism. 24: Glycosaminoglycan degradation. 25: Chemical carcinogenesis. 26: Glucosinolate biosynthesis. 27: Drug metabolism-cytochrome P450. 28: PPAR signaling pathway. 29: Metabolism of xenobiotics by cytochrome P450. 30: Photosynthesis-antenna proteins. * represents genes with significant expression differences.

**Figure 4 plants-14-01001-f004:**
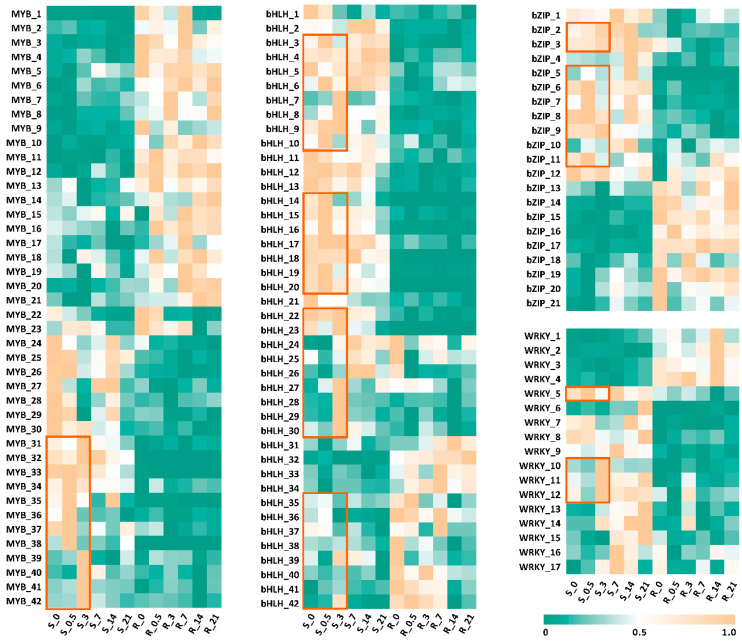
Expression pattern of TF genes including MYB, bHLH, bZIP, and WRKY. R stands for roots, while S stands for stems. The orange box represents genes with increased expression.

**Figure 5 plants-14-01001-f005:**
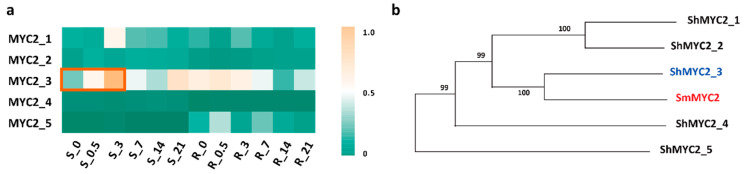
MYC2 in *S. hexandrum.* (**a**) Expression pattern of *ShMYC2*; (**b**) phylogenetic analysis of *SmMYC2* and *ShMYC2s*. R stands for roots, while S stands for stems. The orange box represents genes with increased expression.

**Figure 6 plants-14-01001-f006:**
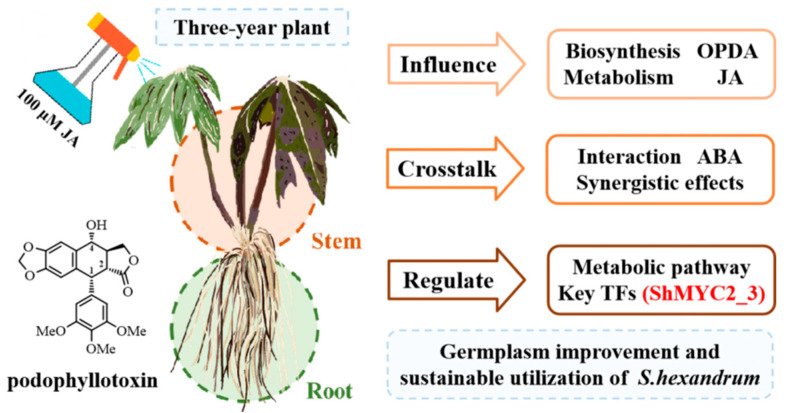
The content and significance of this study.

## Data Availability

The original contributions presented in the study are included in the article/Appendix A; further inquiries can be directed to the corresponding authors.

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
