# Peer review of "The Dynamic Changes in Biosynthesis and Spatiotemporal Distribution of Phytohormones Under Jasmonic Acid Treatment Provide Insights into Hormonal Regulation in Sinopodophyllum hexandrum"

_plants, 2025, doi:10.3390/plants14071001_

Round 1
Reviewer 1 Report
Comments and Suggestions for Authors
Study is focused on changes in biosynthesis and spatiotemporal distribution of endogenous phytohormones under adverse stress in Sinopodophyllum hexandrum. Study is original, provides valuable informations and should be published. However few minor issues should be solved:
In the Materials and Methods chromatografic part of hormone determination is described sufficiently, mass spectrometry is missing almost completely, as well as references to the validation of the method (Chen at al., 2019, Pan et al., 2010).
Author Response
|
Comments 1: In the Materials and Methods chromatographic part of hormone determination is described sufficiently, mass spectrometry is missing almost completely, as well as references to the validation of the method (Chen at al., 2019, Pan et al., 2010).
|
|
Response 1: Thank you for pointing this out. We agree with this comment. Therefore, we have added the mass spectrometry detection methods of phytohormones in 4.Materials and Methods (L418-L421). In addition, we have modified and optimized the logicality and language of the manuscript, hoping to meet your requirements. L418-L421 Mass spectra were acquired in negative-ion mode over a scan range of m/z 50–1500 with a scan time of 0.2 s. The ESI and MS settings were as follows: capillary voltage, 1.0 kV; cone voltage, 40 V; source temperature, 120 °C; desolvation temperature, 450 °C; cone gas flow, 50 L·h-1; desolvation gas flow, 800 L·h-1. Xcalibur was used for data analysis. |
Reviewer 2 Report
Comments and Suggestions for Authors
synthesis of endogenous JA. At this point, it is difficult to make the difference between JA up-taken or neosynthesized. In a general matter, it appears that roots are less receptive to JA production. Because JA administration has been made on leaves, the question arising is: What is measured, is there really endogenous JA biosynthesis or is it the exogenous JA that has been administrated, which is quantified? Is it possible to discriminate between both? The question is yes. A way to do it is to use isotope labeled JA that would be applied as a stress inducer and analyze non-labeled neosynthesized JA form plant extracts. It looks like OPDA production is modified. However, OPDA by itself has other functions than just being a precursor for JA production (see Jimenez Aleman et al., 2022 Phytochemistry volume 204 113432). Nevertheless, it appears that JA decreases quite rapidly, which is correlated with the idea that the phytohormone is degraded. Interestingly, the JA administration has some impact on ABA production even in roots, that means that the signaling is at some extent working in these organs, even if no increase in JA content has been observed (Figure 1b).
In a general matter, I could not understand what is the control and how is it defined when authors say something is increased or not. It is never mentioned what is compared to what? In the M&M section, line 419, there is no information related to this point. Line 347, the way samples were collected is described. The protocol used for JA treatment is quite complex with steps where leaves are sealed in in plastic films for a couple of hours. This protocol can induce stress as well. For this reason, control plants should have been treated as the JA-treated plants, but without the hormone. Analysis of these plants would show whether the JA treatment is significant or not.
Minor remarks:
- In figure 1-2, please indicate what R and S is standing for?
- Line 190: How was gene expression analyzed?
- Line 211: is it not surprising to find streptomycin biosynthesis genes?
- Line 283: operated not operated
- Line 333. “The regulatory effect of ShMYC2_3 on secondary metabolism will be verified by yeast two-hybrid experiment later” What does this sentence mean? How are you going to do that?
- Line 469. The reference is incomplete.
To conclude, the study appears as incomplete with controls that have not been done correctly. Also, it is difficult to find the general biological question and what new elements can answer this question.
Comments on the Quality of English LanguageNo comments.
Author Response
|
Comments 1: Related to synthesis of endogenous JA. At this point, it is difficult to make the difference between JA up-taken or neosynthesized. In a general matter, it appears that roots are less receptive to JA production. Because JA administration has been made on leaves, the question arising is: What is measured, is there real endogenous JA biosynthesis or is it the exogenous JA that has been administrated, which is quantified? Is it possible to discriminate between both? The question is yes. A way to do it is to use isotope labeled JA that would be applied as a stress inducer and analyze non-labeled neosynthesized JA form plant extracts. |
|
Response 1: Thank you for pointing this out. We agree with this comment. We acknowledge that our wording lacked precision, as the levels of JA in plants are not solely determined by endogenous JA biosynthesis. In this study, our primary focus was on examining the dynamic changes in medicinal plants' responses to stress. We have revised the term “endogenous JA content” in the manuscript to “JA content” (L115/L116) and modified the 2.1title “Exogenous JA leads to an increase in endogenous JA levels by directly activating the signaling pathway for endogenous JA biosynthesis in stems” into “Dynamic response characteristics of JA pathway in plant stress regulation” (L105). While JA in this study serves as both an external stress factor and an internal regulatory molecule, it is important to note that JA within the plant indeed plays a role in the dynamic responses and regulatory processes. As for the activation of JA biosynthesis, we have represented the increasing expression levels of the pathway genes in manuscript (L118-L126, Figure 1b). hoping to meet your requirements. L118-L126 By analyzing expression trends of genes involved in JA biosynthetic pathway, we found that in stems, almost all pathway genes showed an upward trend in expression levels shortly after JA application (within 0-3 days). This was especially notable for genes related to the OPDA biosynthesis pathway, including 13-LOX_1/3/4/5, AOS, AOC, and those involved in the conversion of OPDA to JA, such as OPR3, OPCL1 and ACX_2. At the same time, however, the content of JA in roots was too low to be detected, which corresponded to the fact that JA was mainly located in the aboveground part and the most in flowers in the usual cognition. The expression levels of most genes were low in roots before and after JA application (Figure 1b & 1d, Table S3).
|
|
Comments 2: Related to OPDA. It looks like OPDA production is modified. However, OPDA by itself has other functions than just being a precursor for JA production (see Jimenez Aleman et al., 2022 Phytochemistry volume 204 113432). Response 2: We thank the reviewer for this incredible opinion and references provided. By referring to the review provided by the reviewer (Jimenez Aleman et al., 2022 Phytochemistry volume 204 113432), we have learned that OPDA can exert its biological functions independently of the COI1 receptor. This insight helps us better explain the phenomenon of increased expression levels of endogenous OPDA biosynthetic genes but decreased contents, confirming that OPDA plays a crucial role in JA-induced plant responses. Its mechanism of action is likely to be multilayered and involve multiple pathways. Following the reviewer's suggestion, we have revised the relevant content in Section 2.1 (L127-L143) and expanded in 3.Discussion (L296-L306). We sincerely appreciate the reviewer’s key comments and guidance on this study. We have marked relevant contents in red in the manuscript. L127-L143 OPDA is a key signaling molecule in plants response to complex environmental changes, playing a significant biological role in immunity, defense, development, and stress responses. However, when detecting the endogenous OPDA levels, despite significant variation within groups, the overall trend showed a rapid decline in the content of endogenous OPDA (Figure 1a), opposite to the increasing expression levels of OPDA biosynthetic pathway genes. We found that in addition to the CORONATINE INSENSITIVE 1 (COI1) -dependent pathway, OPDA regulates various physiological and developmental processes in plants through multiple mechanisms. For example, OPDA activates signaling pathways by interacting with receptors or channels on the cell membrane, interact with other phytohormones such as auxin and gibberellin, regulate transcription factors, and suppress or activate specific genes[1]. Given the multiple COI1-independent functions of OPDA in plants, we hypothesized that exogenous JA application not only influences the JA-COI1-related pathways but also activates OPDA-COI1-dependent bioactivity. Once OPDA was synthesized, it was likely rapidly metabolized or converted into other bioactive molecules to respond to the plant growth and defense mechanisms triggered by JA, thereby contributing to disease resistance, stress tolerance, and developmental processes, which could lead to a lower accumulation of OPDA in S.hexandrum. L296-L306 The changes in OPDA, an intermediate of JA biosynthesis, showed a complementary pattern to those of JA. Based on the experimental results and theoretical analysis, we speculated that the application of exogenous JA not only activates the JA-COI1-related signaling pathway but also potentially triggers COI1-independent signaling pathways through OPDA. This, in turn, initiates a series of physiological responses, involving plant defense, stress adaptation, immune responses, as well as growth and development. For example, a signal transduction route mediated by OPDA-Ile, which is independent of the SCFCOI1-JAZ co-receptor[2]. Therefore, OPDA plays a crucial role in JA-induced plant responses, and its mechanism is likely to be multilayered and involve multiple pathways, making it an important signaling molecule for plants to adapt to complex environmental changes and stress response.
|
|
Comment 3: Related to JA vanishment. Nevertheless, it appears that JA decreases quite rapidly, which is correlated with the idea that the phytohormone is degraded. Response 3: We thank the reviewer for this comment. As we know, JA transporters and JA inactivators are both key components of the JA signaling pathway, involved in the transport and inactivation of JA, respectively. In Section 2.2, we mainly focused on JA inactivation, including the inactivators JAR1 and the CYP94 family genes (L146-L174, Figure 1d, Table S3). Additionally, following the suggestion of reviewer 3, we searched for JA transporters in the transcriptome, including ABCG transporters, lipid transporters, and MATE transporters. As we primarily focused on the metabolic pathway of JA in section 2.2, we described relevant data in 3.Discussion (L318-L329) and Table S3. Both sections provide evidence for the rapid decline of JA levels in the plant, hoping to meet your approval. We have marked relevant contents in red in the manuscript. L318-L329 JA transporters are also key components of the JA signaling pathway, responsible for the transport of JA across cellular membranes, ensuring its effective distribution throughout the plant and activation of related signaling pathways. These transporters include ATP-binding cassette G-type transporters (ABCG, such as AtABCG16 in Arabidopsis)[3], lipid transporters (LPT, e.g., DIR1[4]), and Multidrug and Toxic compound Extrusion (MATE) transporters[5]. These proteins transfer JA and its derivatives from stimulated cells to others, regulating intracellular JA levels and initiating both local and systemic defense responses. In this study, we also analyzed the expression levels of ABCG, LPT, and MATE in the transcriptome (Table S3) and found that the expression levels of LPTs were specifically increased in response to JA induction, particularly in the roots. This suggested that although JA or its derivatives in the roots were not detected, there was still existing a notable hormonal regulation process and crosstalk occurring in plants.
Comment 4: Related to JA and ABA. Interestingly, the JA administration has some impact on ABA production even in roots, that means that the signaling is at some extent working in these organs, even if no increase in JA content has been observed (Figure 1b). Response 4: We thank the reviewer for this comment. Your suggestion aligns with our intention to highlight the crosstalk between JA and endogenous ABA. Along with the added information on JA transporters, we have further elaborated in 3.Discussion (L318-L329) on the close connection between JA transportation and its crosstalk with other phytohormones. We sincerely thank the reviewer for this insightful comment, greatly enhancing the depth of our study. We have marked relevant contents in red in the manuscript.
Comment 5: In a general matter, I could not understand what is the control and how is it defined when authors say something is increased or not. It is never mentioned what is compared to what? Response 5: We thank the reviewer for this comment. In this study, we focused on the changes in phytohormone content and gene expression levels across different temporal and spatial states, such as comparing the stems and roots at the same time point (Section 2.1, L116-L126), or comparing the same tissue at different time points (Section 2.4, L201-L202). We hope this clarifies your concerns. L116-L126 As shown in Figure 1b, we observed that the JA content in stems rapidly increased within the first 0-0.5 days after exogenous JA application, followed by a gradual decline. By analyzing expression trends of genes involved in JA biosynthetic pathway, we found that in stems, almost all pathway genes showed an upward trend in expression levels shortly after JA application (within 0-3 days). This was especially notable for genes related to the OPDA biosynthesis pathway, including 13-LOX_1/3/4/5, AOS, AOC, and those involved in the conversion of OPDA to JA, such as OPR3, OPCL1 and ACX_2. At the same time, however, the content of JA in roots was too low to be detected, which corresponded to the fact that JA was mainly located in the aboveground part and the most in flowers in the usual cognition. The expression levels of most genes were low in roots before and after JA application (Figure 1b & 1d, Table S3). L201-L202 Given that phytohormone effects are typically short and rapid, we analyzed the differentially expressed genes (DEGs) in stems of JA application at day 0 and day 3.
Comment 6: In the M&M section, line 419, there is no information related to this point. Response 6: We thank the reviewer for this comment. We have removed this sentence. Please forgive our oversight.
Comment 7: Line 347, the way samples were collected is described. Response 7: We thank the reviewer for this comment. We have added additional details about the plant source and treatment methods in section 4.1 (L377-L389), hoping to meet your requirements. We have marked relevant contents in red in the manuscript. L377-L389 Three-year-old S.hexandrum plants grown from seeds from Zhuoni County, Gannan Tibetan Autonomous Prefecture, China (102°46’-104°02’ E, 34°10’-35°10’ N, 2000-3000 meters altitude, temperate continental climate, with an average annual temperature of 8-10 °C, an average relative humidity of 50%, and more than 2000 hours of sunshine per year. The ultraviolet radiation is relatively strong.) were selected and divided into six groups with three replicates in each group, the species was identified by professor Mengfei Li of Gansu Agricultural University. The transparent plastic film was used to cover the cultivation pot of S.hexandrum plants, and 100 μM JA solution, diluted from 2.38 M JA ethanol stock solution, was used to spray the aboveground part of plants on both positive and negative sides of leaves with common spray pot, the droplet size was about 0.035 mm to 0.049 mm, until the solution was dripping on the leaves. Continue spraying 100μM JA solution to form JA water vapor in transparent plastic film. Sealing film for 2 hours. Repeat operation for three times.
Comment 8: The protocol used for JA treatment is quite complex with steps where leaves are sealed in in plastic films for a couple of hours. This protocol can induce stress as well. For this reason, control plants should have been treated as the JA-treated plants, but without the hormone. Analysis of these plants would show whether the JA treatment is significant or not. Response 8: We thank the reviewer for this comment. Same as answer 5, in this study, we focused on the changes in phytohormone content and gene expression levels across different temporal and spatial states, such as comparing the stems and roots at the same time point, or comparing the same tissue at different time points. We hope this clarifies your concerns.
Comment 9: In figure 1-2, please indicate what R and S is standing for? Response 9: We thank the reviewer for this comment. R is standing for roots, while S is standing for stems. We have described this in manuscript (L110 & L195 & L249 & L268). We have marked relevant contents in red in the manuscript.
Comment 10: Line 190: How was gene expression analyzed? Response 10: We thank the reviewer for this comment. We performed second-generation transcriptome sequencing on all plant samples and used the Fragments Per Kilobase of transcript per Million mapped reads (FPKM) method for gene expression normalization to quantify the data. This method allows for the comparison of gene expression levels between different genes or samples. This method has been added in Section 4.5 (L447-L458). We have marked relevant contents in red in the manuscript. L447-L458 4.5. Quantification of Gene Expression Using FPKM in Transcriptome Sequencing In this study, transcriptome sequencing was performed on all plant samples. The gene expression levels were quantified using the Fragments Per Kilobase of transcript per Million mapped reads (FPKM) method. This method normalizes the gene expression data by considering both the length of the gene and the sequencing depth, providing a reliable measure for comparing gene expression levels between different genes or samples. The FPKM calculation is as follows: This standardization method enables accurate comparisons of gene expression across different samples and conditions, accounting for variations in both gene length and sequencing depth. FPKM values were used to assess differential gene expression and provide insight into the gene activity within different experimental treatments.
Comment 11: Line 211: is it not surprising to find streptomycin biosynthesis genes? Response 11: We thank the reviewer for this comment. We speculate that this may be due to microbial symbiosis or contamination. Since S.hexandrum plants were collected from an outdoor environment, their roots or surfaces may have had a symbiotic relationship with microorganisms such as Streptomyces, leading to the detection of streptomycin biosynthesis genes.
Comment 12: Line 283: operated not operated Response 12: We thank the reviewer for this careful observation, and we have revised it according to your kind suggestion.
Comment 13: Line 333. “The regulatory effect of ShMYC2_3 on secondary metabolism will be verified by yeast two-hybrid experiment later” What does this sentence mean? How are you going to do that? Response 13: We would like to thank the reviewer for this comment and apologize for our lack of detailed description. In our subsequent research, the regulatory effect of ShMYC2_3 on secondary metabolism in S. hexandrum will be further verified through yeast two-hybrid experiments. For example, by combining the genes involved in PTOX biosynthesis with ShMYC2_3, we will explore how ShMYC2_3 activates gene expression and regulates metabolic pathways through interactions with target genes, thereby revealing the physical interaction between ShMYC2_3 and the pathway genes. This has also been addressed in the discussion.
Comment 14: Line 469. The reference is incomplete. Response 14: We thank the reviewer for this careful observation, and we have revised it according to your kind suggestion.
Comment 15: To conclude, the study appears as incomplete with controls that have not been done correctly. Also, it is difficult to find the general biological question and what new elements can answer this question. Response 15: We thank the reviewer for this comment. Our primary focus is to investigate the endogenous phytohormone signaling pathways involved in the response and regulation of the rare and endangered medicinal plant S.hexandrum under external stress. Specifically, we aimed to explore the dynamic changes in the biosynthesis and metabolic pathways of JA in S. hexandrum. Especially, we focused on expression levels of genes involved in the JA biosynthetic/metabolic pathways, and the biosynthesis of OPDA, speculating on its subsequent metabolic processes. Additionally, we examined the crosstalk between JA and ABA and identified potential metabolic pathways and transcription factors such as ShMYC2_3. These findings provided valuable insights for understanding the regulation of S. hexandrum secondary metabolism and may serve as a reference for further studies on its metabolic regulation, including for the biosynthesis of lignans like PTOXs. Regarding the reviewer's comment on “controls that have not been done correctly,” as described in our response 5 & 8, our study primarily focused on inter-group comparisons. We aimed to investigate the transcriptional and metabolic changes across different tissues at fixed time points, as well as the temporal variations within the same tissues. This approach was designed to capture the dynamic changes in gene expression and metabolite accumulation in response to JA treatment. We believed this strategy aligns with the goal of identifying key regulatory processes and their contributions to plant stress responses. The primary goal of this study is to provide a reference for the conservation of S. hexandrum—a rare and endangered species from the Tibetan Plateau—and to better understand its response mechanisms to environmental stresses. We hope that these insights will contribute to the preservation of genetic resources and further investigations into its resilience mechanisms in harsh ecological conditions. In addition, we have modified and optimized the logicality and language of the manuscript, hoping to meet your requirements. Please see the attachment |
Reviewer 3 Report
Comments and Suggestions for Authors
I wish to thank the authors for this interesting manuscript. However, I have some comments, please, find them below:
General comments:
1. In this study the effect of JA treatment on JA signalling was investigated. Please, modify the manuscript accordingly (title, Introduction and Discussion).
Information obtained in these experiments can not help to predict how plant would react on the real abiotic stress – heat, salt, drought and so on.
2. Cytokinins and auxins are usually considered as major hormones. In this paper, however, they are ignored completely. Any CK/Aux metabolism related genes were up- or down-regulated in the transcriptome data?
3. Please, check that all used abbreviations were written in full when first used.
4. Figures should be mentioned and discussed consequentially
5. Some references are not complete, so I can not find and verify them.
Some literature is unrelated to the cited context.
Some specific comments:
L119. Why did you make exactly this assumption? For me it looks like the exo-JA was slowly absorbed and metabolised into endo-JA, a rather predictable behaviour.
L121. What about JA transporters and inactivators? Their expression was affected? Please, present the data and discuss.
Section 2.2. Biosynthesis/inactivation rate should be compared. To me, it looks like surplus amount of JA was quickly transported, metabolised and degraded.
L157. Why you speculate when it should be clearly visible from transcriptomics data?
Section 2.4. 1,978 DEGs were mentions, however no genes names in STable 10 – only Cluster.
Based on the presented data, I can not find anything special about MYC2-3.
Also, KEGG “Cutin, suberine and wax biosynthesis” pathway is definitively outperforming. However, it was completely ignored in the results/discussion sections.
L231. There is no ground to suggest that these TF may regulate PTOXs biosynthesis.
L255- 276. These 2 paragraphs just repeat the Introduction, they can be removed.
L277-291. Metabolism should be discussed in full form - biosynthesis, transport, signalling and degradation.
L292-302. Negative regulators/loops also should be discussed based on the data from the Results section.
Section 4.1.
Growth conditions should be specified (humidity, light intensity, temperature and so on)
source for the seeds?
How plants were selected (L348) ?
Spraying system (L353), particularly, the size of the drops? Any additional chemicals added to JA solution?
Section 4.4-4.6.
Please, provide a link to the repository with raw/processed data (NCBI or other).
Also, full transcriptome and DEG data should be provided. In the current form it is just an analysis of several selected TF families which somehow related to the JA.
Author Response
|
Comments 1: In this study the effect of JA treatment on JA signalling was investigated. Please, modify the manuscript accordingly (title, Introduction and Discussion). |
|
Response 1: Thank you for pointing this out. We agree with this comment. After consideration, we believe that exogenous application of JA belongs to stress, and our study was insight into the response mechanism of Sinopodophyllum hexandrum under JA. Therefore, we have changed our manuscript title as “The Dynamic Changes in Biosynthesis and Spatiotemporal Distribution of Phytohormones Under Stress Provide Insights into The Stress Response of Medicinal Plant Sinopodophyllum hexandrum”, and the title of section 2.1 as “Dynamic response characteristics of JA pathway in plant stress regulation”.
|
|
Comments 2: Cytokinins and auxins are usually considered as major hormones. In this paper, however, they are ignored completely. Any CK/Aux metabolism related genes were up- or down-regulated in the transcriptome data? Response 2: We sincerely thank the reviewer for this comment. The changes in secondary metabolites induced by JA are a key focus of our research group. While other hormones also play significant roles in regulating plant development, our primary interest in JA is to provide a foundation for subsequent secondary metabolite measurements.
|
|
Comment 3: Please, check that all used abbreviations were written in full when first used. Response 3: We thank the reviewer for this careful observation, and we have revised it according to your kind suggestion (such as L21/ L26/ L30/ L31/ L132/ L149/ L202). We have marked relevant contents in red in the manuscript.
Comment 4: Figures should be mentioned and discussed consequentially. Response 4: We thank the reviewer for this careful observation. Given that our primary focus is on JA, there may be slight differences in the order of results and figure descriptions. However, we have carefully revised the manuscript to ensure that the text aligns with the figures. We hope this revision meets your expectations.
Comment 5: Some references are not complete, so I can not find and verify them. Some literature is unrelated to the cited context. Response 5: We thank the reviewer for this careful observation, and we have revised it according to your kind suggestion.
Comment 6: L119. Why did you make exactly this assumption? For me it looks like the exo-JA was slowly absorbed and metabolised into endo-JA, a rather predictable behaviour. Response 6: Thank you very much for the reviewer’s professional and insightful comments. We acknowledge that our wording lacked precision, as the levels of JA in plants are not solely determined by endogenous JA biosynthesis. In this study, our primary focus was on examining the dynamic changes in medicinal plants' responses to stress. We have revised the term “endogenous JA content” in the manuscript to “JA content” (L115 & L116) and modified the 2.1title “Exogenous JA leads to an increase in endogenous JA levels by directly activating the signaling pathway for endogenous JA biosynthesis in stems” into “Dynamic response characteristics of JA pathway in plant stress regulation” (L105). While JA in this study serves as both an external stress factor and an internal regulatory molecule, it is important to note that JA within the plant indeed plays a role in the dynamic responses and regulatory processes. As for the activation of JA biosynthesis, we have represented the increasing expression levels of the pathway genes in manuscript (L118-L126, Figure 1b). hoping to meet your requirements. We have marked relevant contents in red in the manuscript. L118-L126 By analyzing expression trends of genes involved in JA biosynthetic pathway, we found that in stems, almost all pathway genes showed an upward trend in expression levels shortly after JA application (within 0-3 days). This was especially notable for genes related to the OPDA biosynthesis pathway, including 13-LOX_1/3/4/5, AOS, AOC, and those involved in the conversion of OPDA to JA, such as OPR3, OPCL1 and ACX_2. At the same time, however, the content of JA in roots was too low to be detected, which corresponded to the fact that JA was mainly located in the aboveground part and the most in flowers in the usual cognition. The expression levels of most genes were low in roots before and after JA application (Figure 1b & 1d, Table S3).
Comment 7: L121. What about JA transporters and inactivators? Their expression was affected? Please, present the data and discuss. Response 7: We thank the reviewer for this comment. As we know, JA transporters and JA inactivators are both key components of the JA signaling pathway, involved in the transport and inactivation of JA, respectively. In Section 2.2, we mainly focused on JA inactivation, including the inactivators JAR1 and the CYP94 family genes (L146-L174, Figure 1d, Table S3). Additionally, following the suggestion of reviewer 3, we searched for JA transporters in the transcriptome, including ABCG transporters, lipid transporters, and MATE transporters. As we primarily focused on the metabolic pathway of JA in section 2.2, we described relevant data in 3.Discussion (L318-L329) and Table S3 to additionally explain JA vanishment. Both sections provide evidence for the rapid decline of JA levels in the plant, hoping to meet your approval. We have marked relevant contents in red in the manuscript. L318-L329 JA transporters are also key components of the JA signaling pathway, responsible for the transport of JA across cellular membranes, ensuring its effective distribution throughout the plant and activation of related signaling pathways. These transporters include ATP-binding cassette G-type transporters (ABCG, such as AtABCG16 in Arabidopsis), lipid transporters (LPT, e.g., DIR1), and Multidrug and Toxic compound Extrusion (MATE) transporters. These proteins transfer JA and its derivatives from stimulated cells to others, regulating intracellular JA levels and initiating both local and systemic defense responses. In this study, we also analyzed the expression levels of ABCG, LPT, and MATE in the transcriptome (Table S3) and found that the expression levels of LPTs were specifically increased in response to JA induction, particularly in the roots. This suggested that although JA or its derivatives in the roots were not detected, there was still existing a notable hormonal regulation process and crosstalk occurring in plants.
Comment 8: Section 2.2. Biosynthesis/inactivation rate should be compared. To me, it looks like surplus amount of JA was quickly transported, metabolized and degraded. Response 8: We thank the reviewer for this comment. We apologized for the limitations of the conditions in this study, which prevent us from supplementing data related to the biosynthesis/inactivation rates, hoping to explore these aspects in future studies. Regarding the rapid decrease in JA content, we have pointed out in section 2.2 that it is associated with the metabolism to JA-Ile and its subsequent rapid oxidation and inactivation (L146-L174). Additionally, in the 3.Discussion section, we have included further descriptions regarding JA transporters (L318-L329), hoping to meet your expectations. We have marked relevant contents in red in the manuscript.
Comment 9: L157. Why you speculate when it should be clearly visible from transcriptomics data? Response 9: We thank the reviewer for this comment. Since we were unable to directly observe changes in content level, we believed that the data analyzed from the transcriptomics provide indirect evidence. Therefore, we used the term "speculate" to express this interpretation.
Comment 10: Section 2.4. 1,978 DEGs were mentions, however no genes names in STable 10 – only Cluster. Response 10: We thank the reviewer for this comment. Since the second-generation transcriptome sequencing only provides annotation information without clear characterization of the genes, we have supplemented the annotation details for these clusters in Table S10, hoping this meets your expectations.
Comment 11: Based on the presented data, I can not find anything special about MYC2-3. Response 11: We thank the reviewer for this comment. According to literatures, we found that four transcription factors—bZIP, MYB, WRKY, and bHLH—are involved in regulating secondary metabolite biosynthesis in two PTOX-producing plants (Kumar, P. et al., 2017 Protoplasma volume 254 1412-1428) (L230-L235). Additionally, the overexpression of SmMYC2 has been shown to significantly induce the phenylalanine and tyrosine biosynthesis pathways(Yang, N. et al., 2017 Frontiers in plant science volume 8 1804) (L252-L255). Based on this information, we identified five ShMYC2 genes in our transcriptome analysis and discovered that the expression pattern of ShMYC2_3 is quite unique. It was significantly upregulated in response to JA treatment. Subsequently, we will further investigate the interaction between this transcription factor and the gene expression products involved in the biosynthesis pathway of triptolide. Therefore, we consider this gene to be of particular interest, and we hope this explanation meets your expectations. We have marked relevant contents in red in the manuscript. L230-L235 We then investigated the most common TFs involved in regulation. Previous studies identified four TFs—bZIP, MYB, WRKY, and bHLH—regulating secondary metabolite biosynthesis in two PTOX-producing plants, S. hexandrum and Podophyllum peltatum. Two unique transcripts encoding bHLH and MYB/SANT TFs were identified in the shoots of P. peltatum, while bZIP and MYB TFs were found in the rhizomes of S. hexandrum, based on differential expression levels. These TFs were closely associated with PTOX content. L252-L255 A previous study found that the biosynthesis of salvianolic acid B, along with the phenylalanine and tyrosine biosynthesis pathways, was significantly induced in 2-month-old transgenic Salvia miltiorrhiza plants that overexpressed SmMYC2.
Comment 12: Also, KEGG “Cutin, suberine and wax biosynthesis” pathway is definitively outperforming. However, it was completely ignored in the results/discussion sections. Response 12: We thank the reviewer for this comment. Cutin, suberine, and wax biosynthesis refer to the processes through which plants synthesize and deposit protective layers on their surfaces. These compounds are crucial for maintaining plant integrity and responding to environmental stresses. These findings provided additional insights into the defense of plants under JA stress, and we have briefly discussed them in section 2.4 (L213-L216). However, since our study primarily focuses on the metabolic regulation of phytohormone levels, we did not elaborate extensively on this aspect. We hope the reviewer understands this point. We have marked relevant contents in red in the manuscript. L213-L216 Additional, cutin, suberine, and wax biosynthesis referred to the processes through which plants synthesize and deposit protective layers on their surfaces. These compounds are crucial for maintaining plant integrity and responding to environmental stresses (Figure 3b, Table S11).
Comment 13: L231. There is no ground to suggest that these TF may regulate PTOXs biosynthesis. Response 13: We thank the reviewer for this comment. Here, we referred to the literature Comparative whole-transcriptome analysis in Podophyllum species identifies key transcription factors contributing to biosynthesis of podophyllotoxin in P. hexandrum.(L230-L235), which suggests that four transcription factors—bZIP, MYB, WRKY, and bHLH—are involved in regulating secondary metabolite biosynthesis in two PTOX-producing plants (Kumar, P. et al., 2017 Protoplasma volume 254 1412-1428). we hope this explanation meets your expectations. L230-L235 We then investigated the most common TFs involved in regulation. Previous studies identified four TFs—bZIP, MYB, WRKY, and bHLH—regulating secondary metabolite biosynthesis in two PTOX-producing plants, S. hexandrum and Podophyllum peltatum. Two unique transcripts encoding bHLH and MYB/SANT TFs were identified in the shoots of P. peltatum, while bZIP and MYB TFs were found in the rhizomes of S. hexandrum, based on differential expression levels. These TFs were closely associated with PTOX content[4].
Comment 14: L255- 276. These 2 paragraphs just repeat the Introduction, they can be removed. Response 14: We thank the reviewer for this careful observation, and we have revised it according to your kind suggestion.
Comment 15: L277-291. Metabolism should be discussed in full form - biosynthesis, transport, signaling and degradation. Response 15: We thank the reviewer for this careful observation, and we have revised it according to your kind suggestion in 3.Discussion (L307-L329). We have marked relevant contents in red in the manuscript. L307-L329 The high expression level of JAR1 may mainly contributes to the rapid decline of endogenous JA in 0.5-3 days. Unfortunately, we were unable to detect JA-Ile in both the roots and the stems. Here, we speculated that the reaction involved in JA-Ile is instantaneous, and it would degrade rapidly after exerting the activity, making the corresponding COI1 quickly returned to the original low expression level. Our first detection time is 0.5-d, at when JA-Ile may had lost its activity by the oxidation activity of CYP94 family genes. Meanwhile, the expression level of CYP94B1, CYP94B3 and CYP94C1_1 did increase in stems. Previous studies have shown that CYP94B3 exerts negative feedback control on JA-Ile levels and performs a key role in attenuation of jasmonate responses[6]. Therefore, we believed that increased expression levels of oxidative metabolism genes were also feedback to maintain the homeostasis of endogenous hormones in S.hexandrum. JA transporters are also key components of the JA signaling pathway, responsible for the transport of JA across cellular membranes, ensuring its effective distribution throughout the plant and activation of related signaling pathways. These transporters include ATP-binding cassette G-type transporters (ABCG, such as AtABCG16 in Arabidopsis)[1], lipid transporters (LPT, e.g., DIR1[2]), and Multidrug and Toxic compound Extrusion (MATE) transporters[3]. These proteins transfer JA and its derivatives from stimulated cells to others, regulating intracellular JA levels and initiating both local and systemic defense responses. In this study, we also analyzed the expression levels of ABCG, LPT, and MATE in the transcriptome (Table S3) and found that the expression levels of LPTs were specifically increased in response to JA induction, particularly in the roots. This suggested that although JA or its derivatives in the roots were not detected, there was still existing a notable hormonal regulation process and crosstalk occurring in plants.
Comment 16: L292-302. Negative regulators/loops also should be discussed based on the data from the Results section. Response 16: We apologize for not considering this aspect during the experimental design. The current data could not provide sufficient evidence to support this point.
Comment 17: Section 4.1.Growth conditions should be specified (humidity, light intensity, temperature and so on), source for the seeds? How plants were selected (L348) ? Spraying system (L353), particularly, the size of the drops? Any additional chemicals added to JA solution? Response 17: We thank the reviewer for this comment. In response to your suggestion, we have modified the Materials and Methods part, including details on the plant source and JA treatment methods (L377-L389). We have marked relevant contents in red in the manuscript. L377-L389 Three-year-old S.hexandrum plants grown from seeds from Zhuoni County, Gannan Tibetan Autonomous Prefecture, China (102°46’-104°02’ E, 34°10’-35°10’ N, 2000-3000 meters altitude, temperate continental climate, with an average annual temperature of 8-10 °C, an average relative humidity of 50%, and more than 2000 hours of sunshine per year. The ultraviolet radiation is relatively strong.) were selected and divided into six groups with three replicates in each group, the species was identified by professor Mengfei Li of Gansu Agricultural University. The transparent plastic film was used to cover the cultivation pot of S.hexandrum plants, and 100 μM JA solution, diluted from 2.38 M JA ethanol stock solution, was used to spray the aboveground part of plants on both positive and negative sides of leaves with common spray pot, the droplet size was about 0.035 mm to 0.049 mm, until the solution was dripping on the leaves. Continue spraying 100μM JA solution to form JA water vapor in transparent plastic film. Sealing film for 2 hours. Repeat operation for three times.
Comment 18: Section 4.4-4.6. Please, provide a link to the repository with raw/processed data (NCBI or other).Also, full transcriptome and DEG data should be provided. In the current form it is just an analysis of several selected TF families which somehow related to the JA. Response 18: We thank the reviewer for this comment. We apologized that this part of the content pertains to a future publication, and the transcriptome data cannot be made publicly available at this time. For DEGs data, we have supplemented the annotation details for these clusters in Table S10, hoping this meets your expectations. Please see the attachment |
Round 2
Reviewer 3 Report
Comments and Suggestions for Authors
Thank you very much for implemented changes.
General comments:
1. Again.
“In this study the effect of JA treatment on JA signalling was investigated. Please, modify the manuscript accordingly (title, Introduction and Discussion).
Information obtained in these experiments can not help to predict how plant would react on the real abiotic stress – heat, salt, drought and so on.”
Stress response was NOT studied in this manuscript. Please, modify sections of the manuscript accordingly (Title, Abstract, Introduction and Discussion). The initial hypothesis should be proven/disproven by conducted experiments, The Introduction section should provide a proper explanation, while The Discussion section – align it with other published research.
In the current version there is no a clear connection between sections.
4. Again.
“Figures should be mentioned and discussed consequentially”.
This comment is still valid.
5. Again
“Some references are not complete, so I can not find and verify them.
Some literature is unrelated to the cited context.”
Please, use reference style recommend by the journal, it includes DOI.
Cited papers should be related to the cited context and support made statements.
Author Response
|
Comments 1: In this study the effect of JA treatment on JA signalling was investigated. Please, modify the manuscript accordingly (title, Introduction and Discussion). Information obtained in these experiments cannot help to predict how plant would react on the real abiotic stress – heat, salt, drought and so on.” Stress response was NOT studied in this manuscript. Please, modify sections of the manuscript accordingly (Title, Abstract, Introduction and Discussion). The initial hypothesis should be proven/disproven by conducted experiments, The Introduction section should provide a proper explanation, while The Discussion section – align it with other published researches. In the current version there is no a clear connection between sections. Response 1: Thank you for your constructive feedback. We have carefully considered your suggestions and made the following revisions to the manuscript:
In response to your comment regarding the study’s focus, we have clarified throughout the manuscript that our work investigates the metabolic and transcriptional responses of plants to JA treatment, rather than studying stress responses induced by abiotic factors such as heat, salt, or drought. The title, abstract, introduction, and discussion sections have been revised to remove any implication that JA represents an abiotic stressor. Instead, we explicitly state that JA treatment is used to simulate stress-like conditions, primarily to determine changes in hormone metabolism and transcriptional regulation in intact plants. Thus, we actively modified the original phrasing of "JA as a stress factor" to "JA treatment" in manuscript and change title into “The Dynamic Changes in Biosynthesis and Spatiotemporal Distribution of Phytohormones Under JA treatment Provide Insights into Hormonal Regulation in Sinopodophyllum hexandrum”. We have made all the necessary revisions, and to ensure clarity, we have highlighted all the changes in yellow within the manuscript.
To address your concerns about the manuscript's coherence: - The 1. Introduction section now includes a more explicit explanation of the study's hypothesis and rationale, supported by relevant references (L65-L74) to demonstrate the hormonal regulatory effects of JA treatment. We emphasize that the aim is to explore metabolic and regulatory responses under JA application, rather than to predict plant behavior under real abiotic stress conditions. L65-L74 (highlighted in yellow) JA treatment can optimize plant physiological performance through complex signaling networks and interactions with other hormones. For example, in Arabidopsis, application of JA can induce the expression of resistance-related genes (such as PDF1.2), thereby enhancing the plant's defense against insects and pathogens[14]. In maize, JA treatment can promote the release of volatile organic compounds (such as terpenoids and green leaf volatiles), attracting natural enemies or parasitoid wasps, and enhancing the plant's indirect defense capabilities[15]. Zhu et al. found that exogenous JA treatment enhances salt tolerance of wheat by regulating the expression of biosynthetic genes for phytohormones such as ABA and salicylic acid (SA)[16]. - The 4. Discussion section has been revised to align our findings with previously published research, highlighting the metabolic and transcriptional responses observed during exogenous JA treatment. We discuss how these findings contribute to understanding the specific signaling pathways activated by JA and their broader implications for plant metabolic regulation. For example, we have added an explanation that JA metabolism is not only related to its activation and subsequent oxidative inactivation, as discussed in the 3. Results section, but also involves transport processes (L309-L320). Additionally, we have included the expression level changes of genes related to JA transport in the transcriptome data in Supplementary Materials, Table S3. L309-L320 (highlighted in yellow) Also, JA transporters are also key components of the JA signaling pathway, responsible for the transport of JA across cellular membranes, ensuring its effective distribution throughout the plant and activation of related signaling pathways. These transporters include ATP-binding cassette G-type transporters (ABCG, such as AtABCG16 in Arabidopsis)[36], lipid transporters (LPT, e.g., DIR1[37]), and Multidrug and Toxic compound Extrusion (MATE) transporters[38]. These proteins transfer JA and its derivatives from stimulated cells to others, regulating intracellular JA levels and initiating both local and systemic defense responses. In this study, we also analyzed the expression levels of ABCG, LPT, and MATE in the transcriptome (Table S3) and found that the expression levels of LPTs were specifically increased in response to JA induction, particularly in the roots. This suggested that although JA or its derivatives in the roots were not detected, there was still existing a notable hormonal regulation process and crosstalk occurring in plants. - We have ensured a clear connection between all sections of the manuscript by consistently framing the study as focused on metabolic changes after JA treatment rather than stress responses.
In response to the Academic Editor's request for clarity, we have provided a detailed description of the JA treatment protocol in Section 4.1. This includes: - A step-by-step outline of the acclimation, treatment, and sampling process for the plants used in the study. L377-L383 (highlighted in yellow) After a 3-day acclimation period, 0-day samples were collected. Subsequently, all aerial parts of plants were treated with 100 μM JA solution, diluted from 2.38 M JA ethanol stock solution, until small droplets formed on the leaves, the droplet size was about 0.035 mm to 0.049 mm. The potted plants were then covered with plastic film to create a sealed environment for 2 hours. The spraying process was repeated on the aerial parts, followed by resealing with the plastic film for three times. After removing the plastic film, samples were collected at various time points. (Figure S1). - An illustrative schematic of the treatment procedure added to the Supplementary Materials (Figure S1, L471). We hope these revisions address your concerns and provide the necessary clarity and coherence to the manuscript. Thank you for your valuable feedback, which has helped us improve the quality of our work. |
|
Comments 2: “Figures should be mentioned and discussed consequentially”. This comment is still valid. Response 2: Thank you for your continued feedback. Our primary focus in this study is JA, and as such, the results section prioritizes the discussion of JA-related findings. While the order of the figures does not always directly correspond to the sequence of the content in the manuscript, we believe that the figures are clearly labeled and described within the text. We are confident that these labels will help readers follow the logical flow of the study without confusion. Moreover, we have made improvements to the order of the tables to ensure they are presented in a more coherent and logical sequence in relation to the text. (Table S6-11, L181, L202, L236, L462, L469, L476-L480) We appreciate your understanding of this issue and hope that these adjustments will still meet the expectations for clarity and structure. Thank you again for your valuable suggestions.
Comments 3: “Some references are not complete, so I cannot find and verify them. Some literature is unrelated to the cited context.” Please, use reference style recommend by the journal, it includes DOI. Cited papers should be related to the cited context and support made statements. Response 3: Thank you for your helpful feedback. We apologize for the incomplete references and any unrelated literature included in the manuscript. In response to your comments: 1. For References Completeness: We have thoroughly reviewed and updated all the references to ensure they are complete and include the necessary DOI information, as recommended by the MDPI reference style. For example, reference 3 (L513), reference 5 (L517), reference 6 (L519). 3. Li, M.; Ge, K.; Sun, T.; Xing, P.; Yang, H. High-elevation cultivation increases anti-cancer podophyllotoxin accumulation in Podophyllum hexandrum. Industrial Crops and Products 2018, 121, 338-344, doi:10.1016/j.indcrop.2018.05.036. 5. Cao, X.; Li, M.; Li, J.; Song, Y.; Zhang, X.; Yang, D.; Li, M.; Wei, J. Co-expression of hydrolase genes improves seed germination of Sinopodophyllum hexandrum. Industrial Crops and Products 2021, 164, doi:10.1016/j.indcrop.2021.113414. 6. Zhao, Q.; Li, M.; Jin, L.; Wei, J. Changes in growth characteristics and secondary metabolites in Sinopodophyllum hexandrum with increasing age. Industrial Crops and Products 2023, 196, 116509, doi:10.1016/j.indcrop.2023.116509.
2. Relevance of Cited Literature: We have carefully checked all cited papers to ensure that they are directly relevant to the context in which they are referenced and support the statements made in the manuscript. Any irrelevant citations have been removed, and we have replaced them with more appropriate references that better align with the study's focus. For example, reference 4 (L515) 4. Shen, S.; Tong, Y.; Luo, Y.; Huang, L.; Gao, W. Biosynthesis, total synthesis, and pharmacological activities of aryltetralin-type lignan podophyllotoxin and its derivatives. Natural product reports 2022, 39, 1856-1875, doi:10.1039/d2np00028h. We hope these revisions meet your expectations and enhance the quality of the manuscript. Thank you for your valuable suggestions. |

Round 3
Reviewer 3 Report
Comments and Suggestions for Authors
I wish to thank authors for implemented changes. The manuscript looks more logical and straightforward now.
As a final comment, please, keep the focus on the hormonal regulation/response.
Keeping in mind that the readers have no access to the transcriptomic data, and that PTOX metabolism was not properly introduced or analysed, or PTOX level measured, It’d be better to remove PTOX from the manuscript, particularly, L98 and the Discussion section (L229-233, 243, 273, 347-355). These sentences do not relate directly to the current manuscript but describe future plans.
Author Response
|
Comments 1: As a final comment, please, keep the focus on the hormonal regulation/response. Keeping in mind that the readers have no access to the transcriptomic data, and that PTOX metabolism was not properly introduced or analysed, or PTOX level measured, It’d be better to remove PTOX from the manuscript, particularly, L98 and the Discussion section (L229-233, 243, 273, 347-355). These sentences do not relate directly to the current manuscript but describe future plans. |
|
Response 1: Thank you for your ongoing support and invaluable guidance throughout this process. Your input has not only refined our article, making it more precise and concise, but also provided us with valuable learning opportunities. Regarding your final comment on focusing more on hormonal regulation/response, we completely agree that the emphasis should remain on this area. It is true that our manuscript initially included some discussion on PTOX, which may not have been fully relevant to the current study, particularly without proper introduction or analysis of PTOX metabolism and its levels. As you pointed out, the focus of our current research is on the hormonal regulation of intact plants. We have carefully considered and addressed these points, removing or revising inferences not supported by actual data while retaining meaningful and relevant discussions. We deleted L98: “and the accumulation of secondary metabolites” L229-233: “These TFs were closely associated with PTOX content” L272-L273: “and more attention has been paid on the change of endogenous secondary metabolite content especially PTOX and flavonoids,” L347-355: “In addition, although the difference in the expression level of phenylpropanoid biosynthesis pathway genes was not as obvious as flavonoids in the results, it still partially indicated that JA may also influenced the content of endogenous lignans.” And we replaced “To provide insights for the subsequent regulation of PTOX biosynthesis” into “To provide insights for transcriptional regulation in JA treatment”. Therefore, we have revised the manuscript to remove references to PTOX and replaced them with "lignans" throughout the text, which are now highlighted in green. This change ensures that the manuscript is more coherent and aligned with the primary objectives of the study. We also appreciate your understanding that our future plans involve further exploration of hormonal regulation in secondary metabolism, especially in relation to lignan biosynthesis especially PTOXs, and we hope to pursue relevant researches in the near future. Once again, thank you for your constructive comments, and we believe these revisions have made the manuscript more focused and clearer. |
